# Biocompatible low-voltage electrothermal actuators with biological operational temperature range
Adéla Slavíková[1], Benjamin C. Baker [1 ✉], Marcos Villeda-Hernandez[1], Annabel Coekin[1], Julia Kwasniewska [1], Tim Good [2], Mina Aleemardani[3], Heidi Snethen[3], James P. K. Armstrong [3] & Charl F. J. Faul [1 ✉]

Muscle loss can severely affect movement and physiological functions, driving interest in artificial muscle development. Although various soft actuators exist, ensuring biocompatibility—especially in terms of heat transfer and non-cytotoxicity—remains a key challenge. To address these issues, here we develop **Bio35**, a low-voltage (3.6 V) electrothermal actuator that operates at mild hyperthermic temperatures (38.9 °C). **Bio35** is synthesized using a one-pot, solvent-free process with Epikote 828, poly(propyleneglycol) bis (2-amino-propyl-ether) (PPG), and 1,4-diamino-diphenyl-sulfone (DDS). It demonstrates high chemical stability, maintaining actuation performance after more than 100 cycles over 200 min. Initial biological tests confirm that these materials are biocompatible and non-cytotoxic. As proof of concept, we demonstrate two systems: a simple gripper capable of holding objects up to 225 mg and a sphincter-like valve, showcasing its potential for use in treating conditions like urinary incontinence, where precise, muscle-like actuation is critical for function.

Skeletal muscles are fundamental to human mobility, allowing us to interact with objects, engage with our surroundings, and navigate our environment[1]. Beyond movement, muscle tissue plays a pivotal role in regulating several internal physiological functions, contributing to homeostasis, health and wellbeing[2]. The magnitude of the force generated by muscles depends on the function of the muscle and the ratio of fast and slow twitch fibers, and can range from a few mN for hair erector muscles to 7 N for sphincter control or 800 N for standing[3,4]. However, muscles are susceptible to damage from various sources, including disease, aging, overuse and traumatic injury[5]. For example, sarcopenia is characterized by an age-related decline in muscle mass and functionality, leading to increased frailty, reduced independence, and a heightened risk of falls and fractures in critical zones like the hip or head[6–8]. Another major, yet underexplored, muscular health issue is female stress urinary incontinence (fSUI), which predominantly affects post-partum females between the ages of 30 and 70, impacting their quality of life significantly[9–11].

Together, these conditions affect around 3% of the global population, representing a major healthcare concern and a significant socioeconomic burden[12]. Despite the considerable impact of these disorders, therapeutic strategies remain limited, necessitating innovative approaches[13]. Attempts have been made to tissue engineer muscle as living grafts, but this approach presents a number of major limitations related to maturation, innervation, and vascularization[14]. In particular, the lack of perfusable vasculature limits the size of the tissue-engineered muscles by restricting the mass transport of oxygen and nutrients[15–17], which in turn affects the overall tissue viability.

In recent years, the emergence of soft and compliant materials within soft robotics has provided promising avenues for creating artificial actuators that are both biocompatible and sustainable[18,19]. Soft robotics is a multi-disciplinary field that integrates electronics[20–22], biology[19,23,24], among others, to develop systems that are functionally analogous to conventional rigid robotics but made entirely from flexible and compliant materials. The mechanical properties, coupled with the ability to undergo large deformations at low supply voltages, have positioned soft material actuators as promising candidates for the development of artificial muscles[25].

Artificial muscles can be designed to respond to various stimuli, including electrical signals[26], thermal changes[27], and pneumatic pressure[4,20]. These stimuli can be used to induce different types of motion as a result of the contraction and relaxation of the materials[28]. Electrothermal actuators have shown significant promise owing to their precise control and rapid responses[29–31]. Thermal expansion mechanisms in electrothermal actuators utilize heat generated by electrical current to produce movement. Soft electrothermal actuators, as a subset of electrothermal actuators, are

[1]School of Chemistry, University of Bristol, Bristol, BS8 1TS, UK. [2]NIHR Long Term Conditions HealthTech Research Centre, Sheffield Teaching Hospitals NHS Foundation Trust, Sheffield, S10 2JF, UK. [3]Department of Translational Health Sciences, Bristol Medical School, University of Bristol, Bristol, BS1 3NY, UK. ✉e-mail: Ben.C.Baker@bristol.ac.uk; Charl.Faul@bristol.ac.uk

typically composed of nonconductive polymers or elastomers that enable significant deformation, and conductive paths that act as resistive heaters[32].

Despite substantial progress in developing electrothermal actuators, several challenges persist. Many existing systems are driven by high voltages, which typically require reinforced insulation for safe operation. Exposure of biological tissue to direct current of just a few hundred microampere can lead to electrochemical injury[33]. Fortunately, electrothermal actuators can be driven with alternating current to permit larger currents within a safe limit in terms of specific absorption rate (SAR) for thermal exposure. In addition, intermittent operation of electrothermal actuators using alternating currents can permit larger currents within a safe limit in terms of SARs for thermal exposure. A temperature of 43.3 °C is considered the maximum temperature that can be applied for several hours in therapeutic settings without irreversible tissue damage[34] or interference with normal cellular functions[35]. Meanwhile, a device activation temperature below body temperature (< 37.2 °C) would result in a permanent physical state (i.e., either thermal expansion or contraction). Taken together, a thermally actuated artificial muscle should ideally be designed with a transition temperature between 37.2 and 43.3 °C. Aside from thermal or electric damage, there are also limitations around the toxicity of the materials currently used as electrothermal actuators. This toxicity is typically manifested by the leaching of substances from the actuator material, which can cause adverse biological reactions in the surrounding tissue (e.g., direct cell damage, systemic toxicity or inflammation[36]). Even though recent advances by Li et al.[37] and Hu et al.[38] have demonstrated significant potential, the biocompatibility and safe temperature ranges (with respect to SAR) remain unaddressed.

There is therefore a pressing need to develop (a) electrothermal actuators that operate within the biological operational temperature range, (b) compact, space-compliant devices with a broader range of operation, avoiding the need for bulky insulation, and (c) systems that are biocompatible and suitable for safe in-vivo use. Our research focuses on the development of a novel bilayer actuator system designed to address these critical issues.

Here we present the development of a highly tuneable epoxy-based bilayer actuator system **Bio35** for artificial muscle applications. By carefully selecting blends of epoxy pre-polymer Epikote 828, 1,4-diaminodiphenylsulfone (DDS), and poly(propylene glycol) bis (2-amino propyl ether) (PPG), and layering them onto Kapton tape, we have developed highly precise (< 1 °C) temperature-responsive actuators capable of significant displacement. It is noted that DDS, under the brand name Dapsone, is currently used as a sulphonamide antibiotic for the treatment of a skin condition, dermatitis herpetiformis, and has been confirmed to be non-toxic. Inclusion of DDS is thus beneficial for the application of these actuators[39].

In this study, we aim to show the influence of thermal expansion within a biological operational temperature range (37.2 to 43.3 °C), test the acute biocompatibility of the material, and explore the cyclability, durability and potential for long-term in vivo applications. We furthermore aim to show our first steps in the development of innovative, biocompatible artificial muscles suitable for the treatment of stress urinary incontinence, sarcopenia, and other muscle-related diseases.

## Results and Discussion
### Actuator formulation
This study presents an optimisation of the composition of epoxy-based bilayer structures to act as efficient actuators at temperatures between 37.2 and 43.3 °C. The active layer was prepared in accordance with Fig. 1a, using blends of Epikote 828, DDS and PPG to give specific thermal expansion

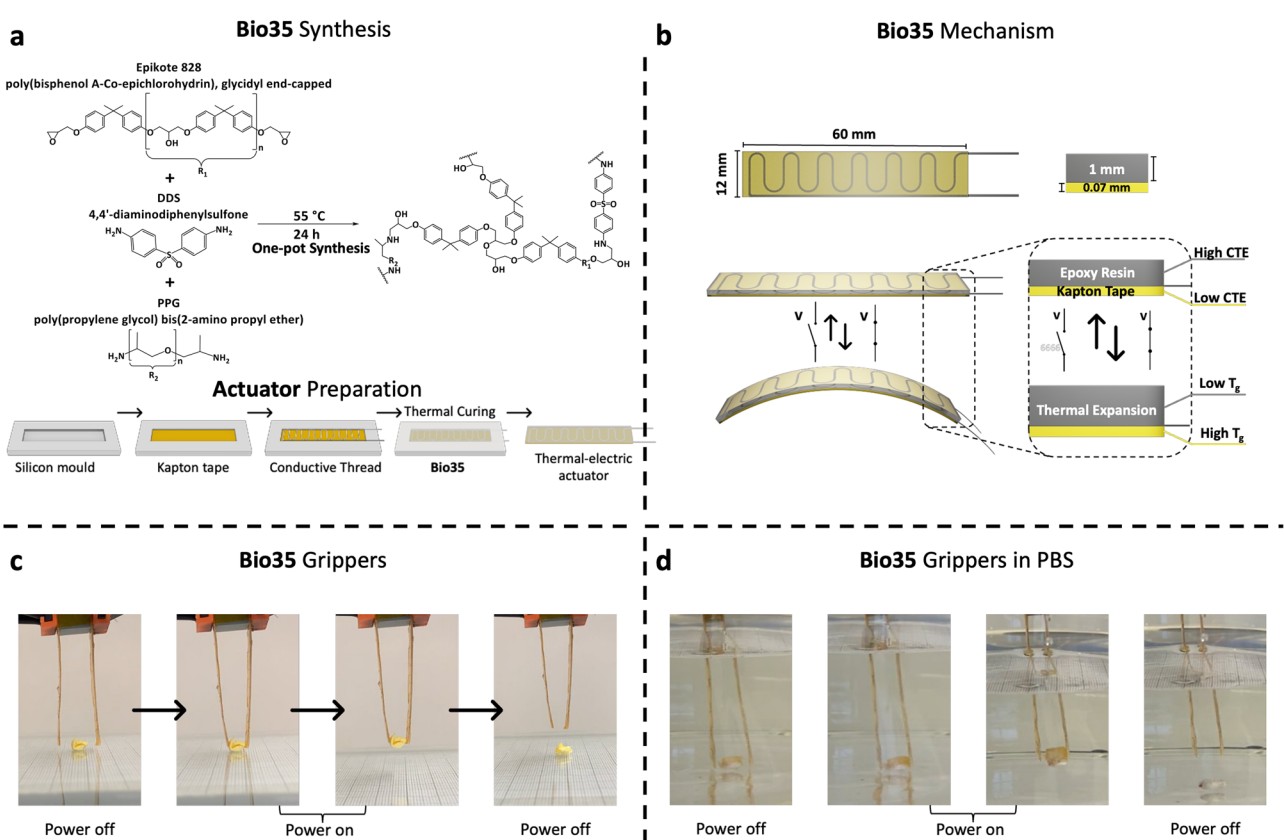

**Fig. 1 | Formulation, manufacturing and application of low voltage and biological operational temperature range bilayer actuators. a** Schematic mechanism of bilayer actuator formation (specifically **Bio35**) from starting materials; **b** configuration of the conductive thread Madeira HC 12 (100% polyamide/silver plated 100 Ω/m) based thermal actuators; **c** demonstration of gripper actuator formed from two bilayer actuators in atmospheric conditions and **d** submerged in phosphate-buffered saline (PBS) solution, where the mass of the actuator is 1 g and the on/off voltage is 5V (200 mA).

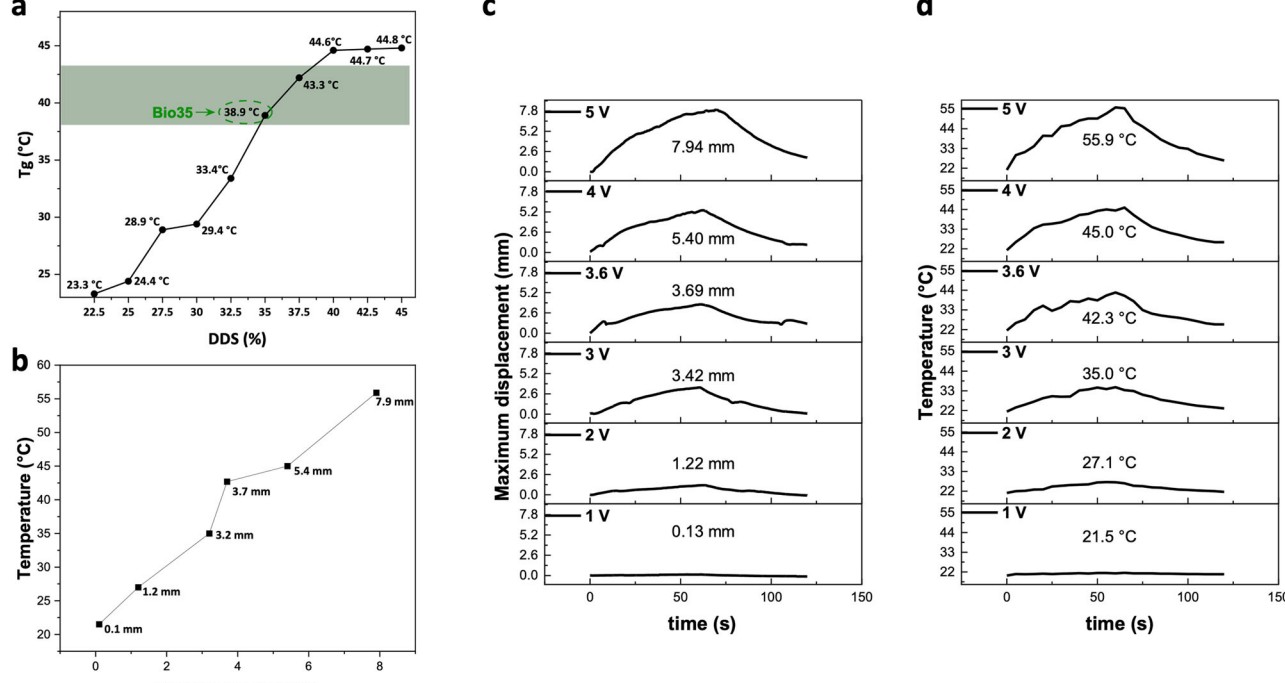

**Fig. 2 | Detailed characterisation of the operating parameters of Bio35 actuator. a** $T_g$ of epoxy cured layers vs percentage DDS used as a crosslinking agent, with the biological operational temperature range indicated by the light green bar; **b** outer temperature profiles of **Bio35** actuator as a function of displacement; **c** maximum displacement of the **Bio35** actuator upon potential application as a function of time; **d** surface temperature as a function of time at different voltages where the actuator dimensions are $0.97 \times 60 \times 12.5$ mm (thickness × length × width). The plots show the average value of 3 repeats on 3 individual samples shown in the **SI**.

properties. It is known from previous studies that differences in the coefficients of thermal expansion (CTE) of layers can be utilised to achieve actuation[27]. Polyimide (Kapton) tape (70 µm thickness) was used as the non-thermally responsive layer to achieve actuation upon heating the active layer (Fig. 1b). Successful formation of crosslinked epoxy resins from the starting materials was confirmed by both FTIR and TGA analysis (see Supporting Information [**SI**] Figure S1 and 2). Conductive silver-coated polyamide thread (Madeira HC 12) was integrated into the active layer to enable Joule heating upon application of an electric current. A simple configuration of two bilayer actuators allowed realisation of a proof-of-concept gripper system capable of lifting 225 mg weight when 5 V (0.6 A) was applied (Fig. 1c). The use of PPG as the polymeric backbone increased the hydrophobicity of the actuator, which allowed a similar operation to be performed in aqueous conditions (Fig. 1d). With a density of less than 1 g cm$^{-3}$, the use of these materials is advantageous in that (gravitational) forces exerted on surrounding tissues will be minimal. In addition, the low volumes required to ensure efficient actuation without additional packaging aids further reduction in unwanted forces that could otherwise lead to tissue damage.

### Actuation mechanism and characterisation

A detailed investigation into tuning the composition of the epoxy-based layer of the bilayer actuator allowed for tailoring the thermal actuation properties of the system. It was previously demonstrated[40] that when the temperature is increased beyond the glass transition temperature ($T_g$) of the material, the rate of expansion significantly increases, thus enabling actuation (Fig. 1b, Fig. S4 and Video S1–2). Differential scanning calorimetry (DSC) analysis was employed to assess the dependence of $T_g$ on the percentage of DDS crosslinker (see Fig. 2a). This analysis showed that the $T_g$ values could be tuned by varying the DDS content, with two formulations (35% and 37.5% DDS) giving $T_g$ values within the biological operational temperature range. It was decided to proceed with investigations using 35% DDS with a $T_g$ of 38.9 °C, as this would allow actuation at mild hyperthermic temperatures. Throughout the text, this actuator composition is referred to

as **Bio35**. Compositions with 37.5% DDS content fell just within the upper limit of the biological operational temperature range, but were not further investigated in this study. However, we propose that these blends could be used in certain operational settings, for example, with greater thermal insulation or in areas that can tolerate greater SAR due to increased cooling effects of higher blood flow.

The bilayer actuators based on **Bio35** could achieve displacements of up to 7.9 mm and a maximum isometric force of 23.91 mN with 5 V applied to the Joule heating system (Fig. S4). The surface temperature of the actuators was monitored throughout actuation, and it was found that application of 3.6 V (corresponding to a displacement of 3.7 mm) was the limit for biologically safe actuation (with a temperature of 42.7 °C reached, see Fig. 2d). The degree of displacement was found to be directly proportional to the voltage applied to the system (see Fig. 2c). The maximum isometric force recorded at 3.6 V was 10.49 mN (although higher forces were achieved at higher voltages, see Fig. S3). Each of these tests were recorded with an actuator of 0.97 mm thickness, 60 mm length, and 12.5 mm width. Exploration of different thicknesses was conducted (see **SI**, Section 2.3), however, it was found that they did not generate the force required or have the response times needed for our proposed applications. The optimal thickness for **Bio35** (0.97 mm thickness, 60 mm length, and 12.5 mm width) generated a power of 18.36 mW with an energy density of 29.7 J g$^{-1}$.

### Actuation optimisation

To assess the applicability of the actuation system for longer-term use, the cyclability of the bilayers was first investigated. Displacement measurements showed that **Bio35** maintained consistent performance over 20 cycles (see Fig. 3a) and over 100 cycles (see **SI**, Fig. S5) using an applied voltage of 3.6 V, without significant loss of displacement (in each case resistance of our actuators was $75 \pm 1 \ \Omega$ m$^{-1}$, monitored over 5 cycles, when in the on state at full displacement). With a typical frequency of 2 min per cycle, our actuator, in this initial study, already showed excellent stability for use over more than 400 min of repeat cycling. Furthermore, temperature monitoring showed

**Fig. 3 | Showing durability of the bilayer actuator Bio35. a** Cyclability in regard to displacement of the actuator against time under atmospheric conditions (25 °C) with 3.6 V applied, and the average total displacement over the cycles shown is 2.91 ± 0.36 mm; **b** Stability of the actuator in aqueous environments with contact angle against time after application of a 2 μL PBS droplet, where the green line shows the **Bio35** DDS layer, the black line shows the **Bio35** Kapton layer and the blue line shows previous epoxy based actuators[27] that have a non-hydrophobic response.

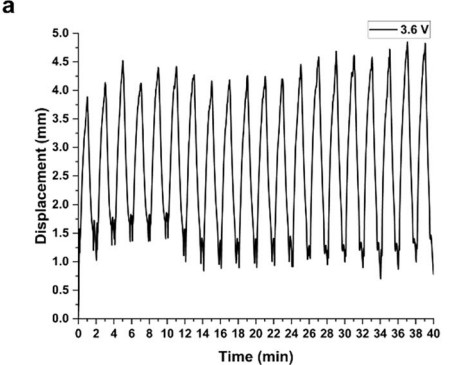

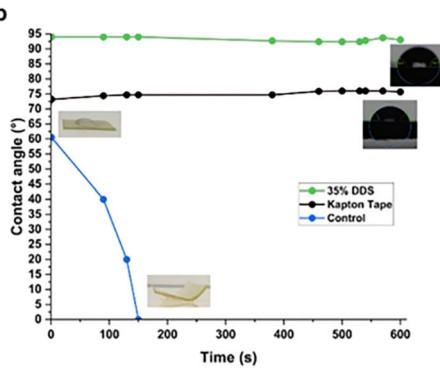

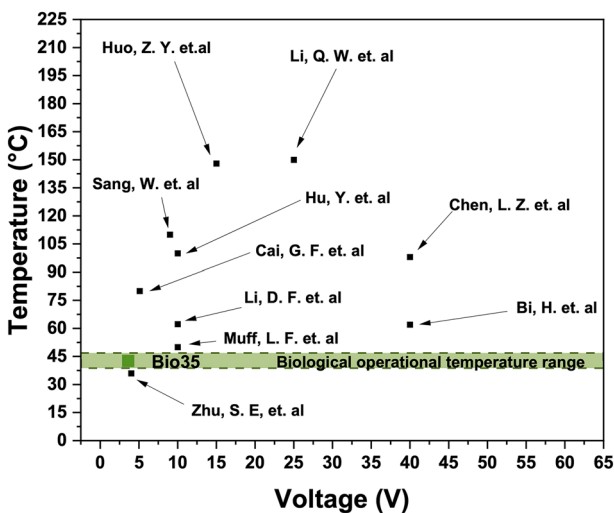

**Fig. 4 | Operating temperature vs driving voltage for recently published Joule-heated soft matter actuators in the field.** The biological operational temperature range is indicated in the green bar[37,38,45–50].

that the actuators stayed within the biologically safe region during cycling. FTIR analysis of the actuator after the cycling test (see **SI**, Fig. S6) indicated no observable material degradation, which was also confirmed by the consistency of the displacement values of 2.91 ± 0.36 mm (Fig. 3a). With respect to a durability timeframe in years, the same actuators were used throughout this study (spanning 2 years) and did not show any decrease in displacement achieved, hydrophobicity or force generated. The response time of the actuators (< 2 s after applied voltage) and the relaxation time (< 30 s after voltage is removed) are shown in Fig. 3 and Fig. S5 and S6.

The stability of **Bio35** after extended contact and exposure to PBS was also assessed. This actuator showed stability in aqueous environments, maintaining its hydrophobicity and flat topology throughout testing. The contact angle remained in the hydrophobic region (93° for the epoxy resin side and 73° for the Kapton tape side for over 10 min). Moreover, unlike previous iterations, the actuator layers were stable against moisture absorption and swelling (see Fig. 3b and SI Table S2).

The durability, cyclability and displacement range of the actuator **Bio35** provide an exceptional opportunity for in vivo applications. **Bio35** significantly outperforms other soft-matter Joule-heated actuators within the field with respect to the applied voltage and outer temperature of the actuator (see Fig. 4). While the excellent work by Zhu et al.[41] provided the first study in the biologically safe temperature range, the displacement of 3.25 μm is over 3 orders of magnitude lower than that achieved by **Bio35**. In addition, unlike **Bio35**, the activation temperature of Zhu's material falls below that of the human core temperature, leaving the actuator permanently in the 'activated' state.

## Testing cytotoxicity of the material

The potential for in vivo applications was further tested by biocompatibility tests using a model cell line (Saos-2). A Transwell assay was performed to assess the cytotoxicity of any components released from **Bio35**. A polycarbonate membrane containing 3 μm pores was used to separate the material and cells while allowing diffusion of chemicals from **Bio35** to the live cultures. An alamarBlue Cell Viability assay was performed on day 1, 7, and 10 of Transwell culture, which revealed no statistically significant difference between the metabolic activity of the cells cultured with or without the material (see Fig. 5a). Next, Saos-2 cells were seeded directly on the surface of **Bio35** and cultured for 24 h. DAPI and phalloidin staining for DNA and F-actin fibres, respectively, followed by confocal fluorescence microscopy revealed a similar number of surface-bound cells to tissue culture plastic controls (see Fig. 5b). The cells bound to **Bio35** presented a typical adherent morphology, characteristic of Saos-2 cells and similar to the cells present on the tissue culture plastic control. With the lack of cytotoxicity, these results represent a significant step towards full biocompatibility (as determined by ISO 10993 standards).

To test the effect of thermally driven actuation on toxicity, cultured Saos-2 cell monolayers in 24-well plates were exposed to a working Bio35 actuator after 24 h. The actuator was immersed in the wells to make direct contact with the cells. Two different voltages (3 V and 5 V) and two cycle counts (5 and 10), with each actuation cycle lasting 1 min of heating 1 min of cooling, were tested. A non-actuated cell monolayer served as the control. Following actuation, alamarBlue analysis revealed no significant difference (n.s.) was observed between the control group and the 3 V, 5-cycle group (Figure S9). However, an 8.7% decrease in cell metabolic activity with increasing voltage and cycle count (5 V, 10 cycles, $p < 0.05$) was observed. These result highlights the validity of our initial approach, while also pointing towards the specific challenges for future development.

## Towards valve applications

To realise the potential for **Bio35** to form sphincter-like replacement implantable valves (as part of a wider approach to address various potential solutions related to fSUI, where 1st and 2nd line treatments such as pelvic floor exercises and bulking agent injection fail) a simple valve using two actuators was prepared (see Fig. 6a). A mock urethral conduit was set up consisting of a reservoir of water (300 mL kept at a constant volume [±0.5 mL] via a drip feed monitored at all times to ensure constant pressure/flow rate) to mimic the bladder and a low-density polyethylene tube (ID = 6 mm, wall thickness = 1 mm, length 60 mm, Fig. 6b) to mimic the urethral conduit (typically with ID = 6 mm, length 40 mm). A voltage of 3.6 V was applied to open the valve for 60 seconds in 10 consecutive intervals to dispense 20 mL of water (Fig. 6c). The artificial sphincter was given two minutes to cool after each cycle. The average rate of flow of water through the tube when the sphincter was open was 1.77 ± 0.10 mL s⁻¹. The average rate of flow through the sphincter after the two minutes of cooling was 0.28 ± 0.26 mL s⁻¹. These results showed that the sphincter was able to

**Fig. 5 | Extended cytoxicity studies of Bio35 actuator. a** Acute cytotoxicity testing was performed using an alamarBlue assay after Saos-2 cells were exposed to Bio35 for 24 h, 7 days, and 10 days. There was no significance (n.s.) in the difference in metabolic activity between the Bio35 group and the control group ($N = 6$) observed in any of the time points. **b** Confocal fluorescence microscopy of Saos-2 cells cultured on Bio35 Actuator for 24 h, fixed and stained for F-actin (phalloidin, green) and nuclei (DAPI, blue). No observable difference was observed in either the number or morphology of cells on the actuator, compared to the control of tissue culture plastic. Scale bar = 200 μm.

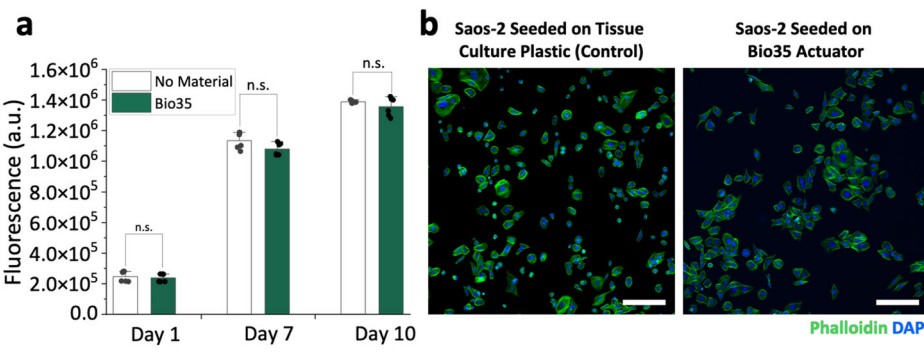

**Fig. 6 | Valve applications of actuator Bio35. a** Creation of the valve actuator from two secured **Bio35** actuators: **b** top-down schematic of the created valve actuator: **c** relative flow rate of water through the artificial urethra tubing (6 mm wide × 0.05 mm thick) through the set up in a when the valve has 3.6 V applied, with average flow rates of $1.77 \pm 0.10$ mL s$^{-1}$ (open flow rate) and $0.28 \pm 0.26$ mL s$^{-1}$ (close flow rate).

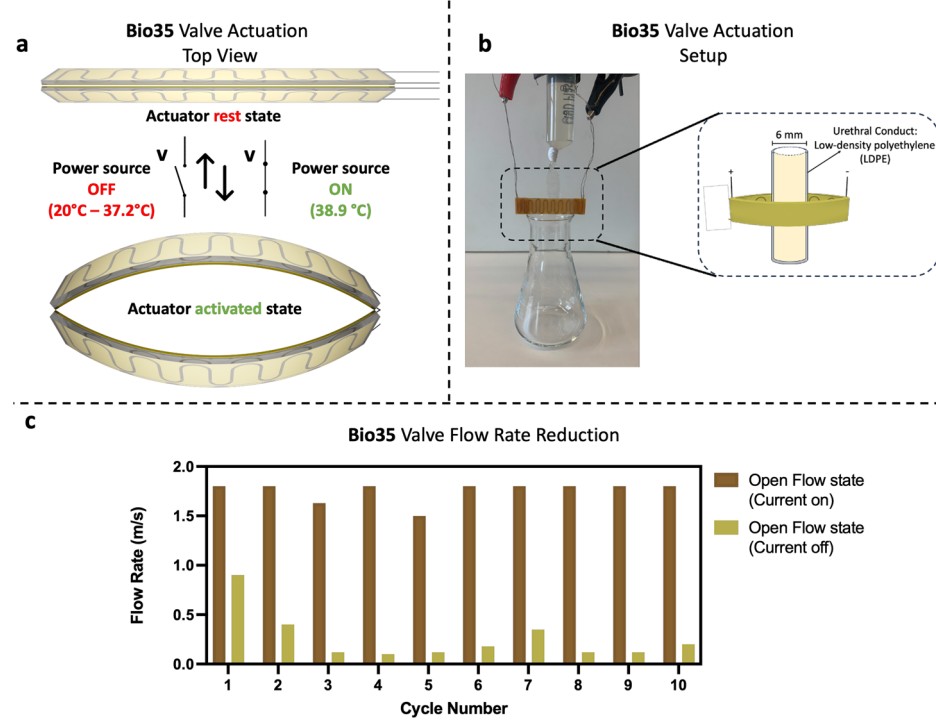

reduce the flow rate by an average of 85%. It is noteworthy that the sphincter was designed to reduce flow significantly, but not entirely shut off all flow. This approach mitigates risk in the proposed fSUI application in which device failure leading to a permanently closed state could cause urinary retention and possible kidney damage during attempted micturition. Furthermore, the ca. 10 mN force presented by the actuators (see Fig. 2) was well below the forces that would cause ischemia by pressure, preventing perfusion into the surrounding tissue. To increase the effectiveness of the actuator it is suggested that implantation will involve a sling like motif (see **SI Fig. S10**).

## Conclusions

In this study we presented results from our investigations into the preparation and optimisation of Joule-heated soft actuators, with the long-term goal of developing fully biocompatible in vivo actuators to aid in addressing the challenge of fSUI. The thermal behaviour of the developed materials could be tuned by simple adjustment of the co-polymerisation ratios. Through simple processing and combination with an in-situ Joule heating configuration, we produced non-cytotoxic actuators that operate in the biological operational temperature range (37.2–42 °C). We could furthermore demonstrate the opening and closing of an artificial sphincter-like

arrangement with a simple valve based on our actuators. The sphincter approach presented here is only a first step towards addressing fSUI. Future work will focus on the development of sling-like implantable actuators to address wider descending bladder and fSUI issues, thus enabling better integration into current surgical procedures for addressing fSUI. In addition, actuators with embedded hyperthermic nanoparticles materials will be explored for the development of remote controlled actuation, thus starting to address the important issues of control and power deliver to these actuators[42–44]. Such solutions will also aim to alleviate the challenges of post-surgical adjustments that are currently unavailable to support patient recovery.

## Methods
### Experimental Methods

**Chemicals.** Chemicals (Merck, UK) were used as received without any purficatioin: poly(bisphenol A – Co – epichlorohydrin) glycidyl end capped, 4,4' diaminodiphenylsulphone, poly(propylene glycol) bis(2-amino propyl ether).

For deionized water a water purifier machine (ELGA PURELAB Ultra GE Genetic MK2, UK) was used. For phosphate buffer solution 800 ml of deionized water, 20.214 g of Sodium Phosphate Dibasic Heptahydrate and

3.394 g of Sodium Phosphate Monobasic Monohydrate was prepared and the solution made up to 1 L with deionized water.

3 M Kapton Tape (Fisher Scientific, UK) with a 0.07 mm thickness and 12.5 mm width. Conductive Thread Madeira HC 12 (Madeira, UK), 100% polyamide/Silver plated <100 ohm m$^{-1}$. Dragon Skin$^{TM}$ 10 VERY FAST (Smooth-On, USA) was used for preparation of silicon form.

**Bio35 Synthesis and Optimisation**. The resin **Bio 35** was synthesized in a simple one-pot synthesis using the prepolymer poly(bisphenol A – Co – epichlorohydrin) glycidyl end capped (Epikote™ 828) and two cross-linkers, 4,4´ diaminodiphenylsulphone (DDS) and poly(propylene gly-col) bis(2-amino propyl ether) (PPG) (see Fig. 1). To ensure all epoxy groups were fully crosslinked a 1:1 molar ratio of resin to crosslinkers was used. The ratio of DDS to PPG crosslinkers was varied to alter the properties of the final resin. The exact amounts of all reagents used can be found in Table S1. All the reagents were mixed at 36 °C for 2 h in a vial using a magnetic stirrer. After this period, the vial was placed in a vacuum oven for 15 min to degas.

**Actuator preparation**. To keep the same sizes of the actuators and ease the manufacturing process, we prepared a negative silicon mold, using commercial polymer Dragon Skin. The negative mold was prepared using a 3D printed resin from an SLA printer. The Kapton Tape (0.07 mm thickness, 12.5 mm width) was placed into the silicon mold, sticky side up. To introduce the joule heater a silver conductive thread was placed onto the Kapton Tape layer arranged into a desired shape. Finally, the synthesized resin was poured on the tape with the thread, and the mold with the actuator was placed into an oven for curing for 24 h at 55 °C.

**Actuator Molds**. The design of the negative molds was carried out using Fusion 360 (Autodesk, USA) and subsequently printed using a Mono X 6 K SLA 3D printer (Anycubic, CN) with a high transparency resin. After printed, the molds were thoroughly washed with isopropyl alcohol (> 99% Sigma-Aldrich, DE) and dried to remove uncured resin. A post-curing process was conducted for 20 s in a custom-built UV reactor comprising 4 UV LEDs emitting a wavelength of 365 nm (UVA) and a 1 W power output each. To prevent the inhibition of the curing process of the silicone rubber, the molds were coated with a polyurethane varnish. Positive molds were then created using Dragon Skin silicone rubber (Smooth-On, USA); the components of Dragon Skin A and B were premixed, poured into the prepared molds, and allowed to cure for 30 min.

**Gripper actuator**. As one of the proofs of concept, we prepared a gripper. Two of the **Bio35** actuators were joined together, Kapton layer inside, using another Kapton Tape. The voltage was applied (Aim-TTi EL155R Digital Bench Power Supply, UK) to a gripper to grab the paper ball weight (weighing 225.1 mg) and hold it untill the voltage was cut off.

**Valve actuator**. As a proof-of-concept, we prepared a valve. Two **Bio35** actuators were stuck together at the ends, using another Kapton Tape, to form a sphincter, Kapton Tapes inside. After applying the voltage, the sphincter is opening and releasing the water ("urine") from the syringe ("bladder").

**Actuation**. The actuator was clipped to a stand. The source (Aim-TTi EL155R Digital Bench Power Supply, UK) was joined to the silver thread ends by crocodile clips and graph paper (1 mm squares) was used for measuring the displacement. The applied voltages were in intervals from 1 to 5 V. Before every measurement, 3 preheating cycles were run.

**Measuring Rate of Flow**. To quantify the effectiveness of the sphincter design at restricting urine flow a bespoke setup, shown in Fig. 5, was designed to simulate urination. A thin tube (Low-density Polyethylene, OD 6 mm) representing the urethra was attached using vacuum thread to a 20 ml syringe tube which represented the bladder. The average female urethra is 6 mm wide, so tubing as near to 6 mm as possible was used. The sphincter was placed around the plastic tube. The bladder was kept constantly full of 20 ml by manually topping up and the water allowed to dissipate under gravity.

**Characterization**. Fourier-transform infrared spectroscopy (FTIR): The FTIR measurements were run on a spectrometer with a universal ATR two modular accessory with a diamond crystal (PerkinElmer Spectrum 100, USA). Sample spectra were recorded at wavenumbers between 450 and 4000 cm$^{-1}$ at a resolution of 4 cm$^{-1}$ with 15 scans.

Thermogravimetric analysis: Thermal stability of the formulations of interest DDS 35% (Bio35) and 37.5% was studied using a simultaneous thermal analysis instrument (Netzsch STA 449 F1 Jupiter, Selb, Germany); thermogravimetric analysis (TGA) data was obtained. The cured resins (ca. 10 mg) were added to Al$_2$O$_3$ crucibles and subjected to a temperature programme from 30 °C to 800 °C at 5 °C min$^{-1}$, under Nitrogen atmosphere (50 cm$^3$ min$^{-1}$).

Differential Scanning Calorimetry (DSC): The DSC measurements were taken on a DSC25 (TA instruments, UK), the T$_g$ was then analysed using Trios software as an inflection temperature in the DSC curve. A heating and cooling rate of 10 °C min$^{-1}$ was used in a temperature range of -50 °C to 100 °C. Samples were analysed in Tzero aluminium pans with hermetic lids.

Force measuring: The force measurements were conducted using a load cell (RB-Phi-203-100g Micro Load Cell, RoboShop, UK) connected to a data acquisition system (DAQ) (NI 779051-01). The load cell outputs were digitized by the DAQ, enabling accurate retrieval and processing of the data. The DAQ system was interfaced with a computer running MATLAB software, which was used for logging, analysing, and visualizing the data accordingly. The actuator was mounted in an uniaxial support, with one end firmly clipped to the fixed side of the support. The free end of the actuator was positioned and aligned in close proximity to the load cell to ensure accurate force measurements at the point of contact.

Testing of cyclability: A laser (Keyence laser LK-G157, UK) was used to measure 200 cycles. The laser was mounted on a height adjustable stand and could also be rotated for different measurement angles. The laser was connected to a laptop with LK navigator software. The following settings were sent from the software to the laser: 1 ms measurements, data points recorded every 100th measurement, for 65,000 data points. The actuator was positioned using a clamp stand horizontal and perpendicular to the laser beam. The laser was pointed at the actuator and the distance between the device and the actuator adjusted until the light on the device head flashed green. The device could then be zeroed, and data recording could begin. One cycle lasts 2 min–1 min heating, 1 min cooling down. The temperature of the actuator was measured every 5th cycle.

Contact angle measuring: Contact angle measurements were taken on a DSA100 (KRÜSS, UK), with 2 µl of PBS droplets. The measurements were conducted at a temperature of 20 ± 0.5 °C. The droplet images were captured across 600 s.

Cell culture: Saos-2 cells (89050205, Merck) were thawed from liquid nitrogen storage and cultured in McCoy's 5 A medium (HyCloneTM, UK) with 10% fetal bovine serum (v/v) (Thermo Fisher Scientific, UK), 100 units mL$^{-1}$ penicillin (Thermo Fisher Scientific, UK), and 100 µg mL$^{-1}$ streptomycin (Thermo Fisher Scientific, UK), and incubated at 37 °C with 5% CO$_2$. Once the cells reached 80% confluency, they were washed with Dulbecco's phosphate-buffered saline (DPBS, Thermo Fisher Scientific) and treated with 5 mL of trypsin-EDTA solution (Sigma-Aldrich, UK) for 3 min. The cells were then counted using a haemocytometer before being seeded (5 × 10$^3$ cells per well) onto the 24-well plates. **Bio35** samples (N = 6) were placed into transwell inserts for 24-well plates (PET, translucent, 3 µm pore size; Sarstedt, Germany) and then added to the 24-well plates seeded with cells. A total of 500 µL of media was added to each well. Also, as a control, Saos-2 cells were seeded directly on 24 well-plates without any samples (no material, N = 6).

alamarBlue Assay: The metabolic activity of Saos-2 cells was assessed using the alamarBlue assay (Thermo Fisher Scientific). The alamarBlue stock solution was mixed with the cell culture medium in a 1:10 ratio to create the alamarBlue/medium solution. On days 1, 7, and 10 of culture, 500 μL of this solution was added to each well and incubated for 2 h. Subsequently, 100 μL from each well was transferred to a 96-well plate in triplicate. Readings were taken using a microplate reader (SYNERGY-H1, BioTek Instruments, Inc.) set to excitation and emission wavelengths of 560 nm and 590 nm, respectively.

For the effect of actuation: A total of $1 \times 10^5$ cells were seeded per well onto 24-well plates and cultured for 24 h in an incubator at 37 °C with 5% $CO_2$. Actuation was then applied to the seeded Saos-2 cells using the Bio35 actuator, powered by an EL 155 R power supply, at 3 V (5 and 10 cycles) and 5 V (5 and 10 cycles), with each cycle lasting 1 min ($N = 3$). During actuation, the Bio35 actuator was immersed in the well to make contact with the cell monolayer and assess its effect on the cells. Following actuation, cell metabolic activity was evaluated using the alamarBlue assay, as described previously. Seeded cells on tissue culture plastic with no actuation served as the control ($N = 3$).

Confocal fluorescence microscopy: The **Bio35** samples were sterilised with three washes in 70% ethanol for 10 min each, followed by three washes in sterile Dulbecco's phosphate buffered saline (PBS) for 10 min each. The **Bio35** samples were then soaked in cell culture media for one day, before seeding Saos-2 cells at a density of $5 \times 10^3$ cells per sample, in 25 μL of cell suspension. The cell-seeded samples were incubated for 1 h at 37 °C with 5% $CO_2$, before adding 1 mL of cell culture media. Tissue culture plastic (TCP) was seeded in the same way and used as a control. After 24 h of culture, the samples were fixed in 3.7% formaldehyde (Sigma-Aldrich, UK) for 20 min, gently washed with PBS, permeabilised in 0.1% (v/v) Triton X-100 (in PBS, Sigma-Aldrich, UK) for 20 min, then washed three times with PBS. F-actin filaments were stained by incubating with a 4 U mL$^{-1}$ solution of phalloidin conjugate (FITC, Thermo Fisher Scientific, UK) in the dark for 30 min. Samples were washed with PBS, then cell nuclei were stained by incubating with a 1 μg mL$^{-1}$ solution of 4′,6-diamidino-2-phenylindole (DAPI, Thermo Fisher Scientific, UK) in the dark for 15 min. The samples were then washed with PBS and imaged with a Leica SP8 confocal fluorescence microscope using a 110x HC PL APO CS2 objective. DAPI was imaged using a 405 nm laser excitation and detected in the 415–480 nm range. FITC was imaged using a 488 nm laser excitation and detected in the 498–550 nm range. Z-stack images (1024 × 1024 pixels) were captured for each sample and then combined into a single image using maximum projection in LAS X software.

## Data availability
All data supporting the findings of this study are available within the article and its Supplementary Information files. Additional data are available from the corresponding author upon reasonable request.

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

## Acknowledgements

B.C.B., M.V.H., T.G., M.A., J.P.K.A., and C.F.J.F. acknowledge EPSRC EP/R511663/1. J.P.K.A. acknowledges funding from a UKRI Future Leaders Fellowship (MR/V024965/1).

## Author contributions

A.S., B.C.B., and M.V.H. contributed equally to this work. A.S.—lab work and writing, B.C.B – lab work, supervision, writing, conceptualisation, M.V.H.—lab work, supervision, writing, conceptualisation, A.C.—Lab work, J.K.—Lab work, T.G.—Advisory, writing, M.A. – Lab work and writing, H.S.—Lab work, J.P.K.A.—supervision, writing, C.F.J.F—supervision, writing, conceptualisation.

## Competing interests
The authors declare no competing interests.
