## [Transparent Peer Review file · Communications Materials]

Towards Muscle Replacement: Biocompatible Low-Voltage Electrothermal Actuators

Corresponding Author: Dr Benjamin Baker

Version 0:

Decision Letter:

Dear Dr Baker,

Thank you for submitting your manuscript, "Towards Muscle Replacement: Biocompatible Low-Voltage Electrothermal Actuators", to Communications Materials. It has now been seen by 3 referees, whose comments are appended below. You will see that while they find your work of potential interest, they have raised substantial concerns that must be addressed. In light of these comments, we cannot accept the manuscript for publication, but are interested in considering a revised version that addresses these serious concerns.

In particular, Reviewer 2 requests additional experiments in order to verify the real applicability of the device.

We hope you will find the referees' comments useful as you decide how to proceed. Should further experimental data or analysis allow you to address these criticisms, we would be happy to look at a substantially revised manuscript. However, please bear in mind that we will be reluctant to approach the referees again in the absence of major revisions. If the revision process takes significantly longer than three months, we will be happy to reconsider your paper at a later date, as long as nothing similar has been accepted for publication at Communications Materials or published elsewhere in the meantime.

When submitting your revised manuscript, please include the following:

-A response letter with a point-by-point reply to each of the referee comments and a description of changes made. Please include the complete referee report in the response letter. Please note that the response letter must be separate to the cover letter to the editors.

-A marked-up version of the manuscript with all changes to the text in a different colored font. Please do not include tracked changes or comments. Please select the file type 'Revised Manuscript - Marked Up' when uploading the manuscript file to our online system.

-A clean version of the manuscript. Please select the file type 'Article File'.

-An updated <https://www.nature.com/documents/nr-editorial-policy-checklist.zip> Editorial Policy checklist, uploaded as a 'Related Manuscript File' type. This checklist is to ensure your paper complies with all relevant editorial policies. If needed, please revise your manuscript in response to these points. Please note that this form is a dynamic 'smart pdf' and must therefore be downloaded and completed in Adobe Reader. Clicking this link will download a zip file containing the pdf.

Please use the following link to submit your revised manuscript files:

Link Redacted

Please do not hesitate to contact me if you have any questions or would like to discuss the required revisions further. Thank you for the opportunity to review your work.

Best regards,

Maria Rosa Antognazza, PhD
Editorial Board Member
Communications Materials
orcid.org/0000-0003-4599-2384

Reviewers' comments:

Reviewer #1 (Remarks to the Author):

The paper introduces the concept of biocompatible low-voltage electrothermal actuators and presents Bio35, a specific low-voltage (3.6 V) actuator along with experimental results. While the topic is indeed relevant and interesting to the field, there are several significant issues that, in my view, prevent the manuscript from meeting the rigorous standards of Communications Materials. To improve the paper, I offer the following suggestions:

1. The manuscript is poorly organized and lacks clarity. The introduction section is excessively long and disproportionate to the rest of the manuscript. A concise and focused introduction is crucial for setting the stage for the research, but this section contains too much background information and tangential details.
2. The figures in the manuscript are of poor quality, with text and labels that are too small to be legible. The authors should ensure that all figures are of high resolution and that all text, labels, and axes are sufficiently large to be easily read by the journal's target audience.
3. The format of the figure captions in the manuscript is inconsistent, which detracts from the professional appearance and readability of the article.
4. The references in the manuscript do not adhere to the journal's formatting guidelines.

For these reasons, I believe the paper is not yet ready for publication in Communications Materials.

Reviewer #2 (Remarks to the Author):

The manuscript presents an epoxy-based electrothermal actuator designed as a potential replacement for defective muscle caused by aging or disease. The proposed material utilizes a design strategy originating from work published in 1989, with modifications to optimize actuation under conditions relevant to biological applications. The study also includes cytotoxicity assessments and evaluates the actuator's potential to function as a valve for managing urinary incontinence using a mock urethral conduit. While the concept of employing a synthetic actuator for biological applications is intriguing, several key issues require further attention to improve the rigor and applicability of the work:

1. The manuscript highlights the actuator's potential for managing urinary incontinence but overlooks alternative approaches that are already in use, such as pharmacological treatments, physical therapy, and injectable bulking agents. The introduction would benefit from a broader discussion of these strategies to better contextualize the proposed actuator's niche or advantages.
2. A critical consideration for implantation is the mechanism by which the actuator would function autonomously, given that natural valves operate under neural control. Additionally, practical details about the source and delivery of electrical power to the actuator need to be addressed.
3. Actuator performance was evaluated over a relatively short duration (40 minutes), which is insufficient for applications requiring long-term reliability. Extended testing is necessary to assess durability and functional stability over a biologically relevant timeframe.
4. Figures and data presented in the manuscript should clarify the number of independent samples tested under each condition. If the results are based on a single sample, this should be explicitly stated, and the limitations should be acknowledged.
5. Although the authors argue that the working temperature of 42.7°C is safe, prolonged exposure to this level of heat can induce oxidative stress and structural damage in surrounding tissues. Additional experiments are needed to evaluate the effects of long-term heat exposure on adjacent biological structures.
6. The cytotoxicity assessment was conducted over a relatively short duration (24 hrs). Testing over a longer period (e.g., at least one week) is necessary to account for potential material degradation and the release of toxic substances in a physiological environment.

Reviewer #3 (Remarks to the Author):

This paper propose a bilayer electrothermal actuator for potential applications in urinary sphincter replacements. The produced non-cytotoxic actuators can operate under low voltage and in the biological operational temperature range (37.2-42°C). The materials are also proved biocompatible. The actuation performance and durability are also tested. Finally, a simple valve actuator is demonstrated to mimic the opening and closing of an artificial sphincter. Main comments are as follows:

1. The font size in figures seems too small.
2. I don't think only 100 cycles can prove the good durability of the actuators, especially for potential biocompatible applications, the working environments (such as temperature, humidity, PH, etc.) can affect the applicability and durability. Please add some discussion about this.
3. What's the resistance of this actuator? During actuation does the resistance vary with time or voltage?
4. What's the power or energy density of this actuator?
5. For the valve application, actually the pressure in the reservoir has significant effect on the flow rate and also the actuation of the valve actuator, because we know the urinary bladder is elastic. The demonstration only show the free flow of the water by gravity. The applicability of this valve actuator is not convincing.
6. The actuation strain and force are both very small compared to many other electrothermal actuators, please give some reason or discuss about some potential strategy for improvement.

Communications Materials is committed to improving transparency in authorship. As part of our efforts in this direction, we are now requesting that all authors identified as 'corresponding author' create and link their Open Researcher and Contributor Identifier (ORCID) with their account on the Manuscript Tracking System prior to acceptance. ORCID helps the scientific community achieve unambiguous attribution of all scholarly contributions. You can create and link your ORCID from the home page of the Manuscript Tracking System by clicking on 'Modify my Springer Nature account' and following the instructions in the link below. Please also inform all co-authors that they can add their ORCID to their accounts and that they must do so prior to acceptance.

Version 1:

Decision Letter:

Dear Dr Baker,

Thank you for submitting your manuscript, "Towards Muscle Replacement: Biocompatible Low-Voltage Electrothermal Actuators", to Communications Materials. It has now been seen again by 3 referees, whose comments are appended below. You will see that while they find your work of interest, some important points are still raised. We remain interested in the possibility of publishing your study in Communications Materials, but would like to consider your response to these concerns in the form of a revised manuscript before we make a decision on publication.

We therefore invite you to revise and resubmit your manuscript, taking into account the points raised.

When submitting your revised manuscript, please include the following:

-A response letter with a point-by-point reply to each of the referee comments and a description of changes made. Please include the complete referee report in the response letter. Please note that the response letter must be separate to the cover letter to the editors.

-A marked-up version of the manuscript with all changes to the text in a different colored font. Please do not include tracked changes or comments. Please select the file type 'Revised Manuscript - Marked Up' when uploading the manuscript file to our online system.

-A clean version of the manuscript. Please select the file type 'Article File'.

-An updated <https://www.nature.com/documents/nr-editorial-policy-checklist.zip> Editorial Policy checklist, uploaded as a 'Related Manuscript File' type. This checklist is to ensure your paper complies with all relevant editorial policies. If needed, please revise your manuscript in response to these points. Please note that this form is a dynamic 'smart pdf' and must therefore be downloaded and completed in Adobe Reader. Clicking this link will download a zip file containing the pdf.

In the event that your manuscript is accepted we will provide detailed guidance on our journal policies and formatting. You may however wish to ensure that the manuscript complies with our house style at this stage. See our style and formatting guide (<https://www.nature.com/documents/commsj-phys-style-formatting-guide-accept.pdf>) and checklist (<https://www.nature.com/documents/commsj-phys-style-formatting-checklist-article.pdf>) for reference.

Data availability statements and data citations policy: All Communications Materials manuscripts must include a section titled "Data Availability" at the end of the Methods section or main text (if no Methods). More information on this policy, and a list of examples, is available at <http://www.nature.com/authors/policies/data/data-availability-statements-data-citations.pdf>.

- Accession codes for deposited data
- Other unique identifiers (such as DOIs and hyperlinks for any other datasets)
- At a minimum, a statement confirming that all relevant data are available from the authors
- If applicable, a statement regarding data available with restrictions
- If a dataset has a Digital Object Identifier (DOI) as its unique identifier, we strongly encourage including this in the Reference list and citing the dataset in the Data Availability Statement.

DATA SOURCES: We strongly encourage authors to deposit all new data associated with the paper in a persistent repository where they can be freely and enduringly accessed. We recommend submitting the data to discipline-specific, community-recognized repositories, where possible and a list of recommended repositories is provided at <http://www.nature.com/sdata/policies/repositories>.

If a community resource is unavailable, data can be submitted to generalist repositories such as <https://figshare.com/> or <http://datadryad.org/> Dryad Digital Repository. Please provide a unique identifier for the data (for example a DOI or a permanent URL) in the data availability statement, if possible. If the repository does not provide identifiers, we encourage authors to supply the search terms that will return the data. For data that have been obtained from publically available sources, please provide a URL and the specific data product name in the data availability statement. Data with a DOI should be further cited in the methods reference section.

Please use the following link to submit your documents:

Link Redacted

We hope to receive your revised paper within six weeks; please let us know if you aren't able to submit it within this time so that we can discuss how best to proceed. If we don't hear from you, and the revision process takes significantly longer, we will close your file. In this event, we will still be happy to reconsider your paper at a later date, as long as nothing similar has been accepted for publication at Communications Materials or published elsewhere in the meantime.

Please do not hesitate to contact me if you have any questions or would like to discuss these revisions further. We look forward to seeing the revised manuscript and thank you for the opportunity to review your work.

Best regards,

Maria Rosa Antognazza, PhD
Editorial Board Member
Communications Materials
orcid.org/0000-0003-4599-2384

Reviewers' comments:

Reviewer #1 (Remarks to the Author):

The authors have carefully addressed the review comments. But I think the current version is not suitable for acceptance. Below are detailed comments to guide revisions.

1. Compare and contrast the proposed Bio35 actuator with existing approaches to clearly highlight its novelty and advantages.
2. Improve the quality and clarity of the figures. Consider using color coding, annotations, and captions to make the figures more self-explanatory.
3. Although the manuscript details the synthesis process of Bio35, I find that the clarity of Figure 1a remains inadequate for precise interpretation.
4. The camera perspectives in Figures 1c and 1d appear inconsistent, which may affect the reproducibility and interpretability of the experimental setup depicted.
5. The numerical labels and annotations in Figure 5a are too small.
6. The schematic presented in Figure 6a does not adequately depict the electromechanical actuation process of the valve actuator.
7. The manuscript presents data on the displacement and force generated by Bio35 actuators at 3.6 V. However, it would be useful to include data on the response time of the actuators and the relaxation time.
8. It would be helpful to provide more information about the design of the mock urethral conduit, the specific parameters used during the valve opening and closing tests (e.g., flow rate, pressure).
9. Extend the duration and scope of the biocompatibility tests to ensure the long-term safety and stability of the Bio35 material in vivo.
10. The references section is comprehensive, but some key citations could be updated to include more recent studies on artificial muscles and soft robotics.

Reviewer #2 (Remarks to the Author):

The authors made sincere efforts to respond to the original comments. In my opinion, the manuscript is quite qualified for acceptance.

However, there is one suggestion on the response to "A critical consideration for implantation is the mechanism by which the actuator would function autonomously, given that natural valves operate under neural control. Additionally, practical details about the source and delivery of electrical power to the actuator need to be addressed." The proposed hyperthermia nanoparticles as a power delivery measure are speculative. If particular published works support this statement, authors should cite the papers. Or, authors should elaborate on how nanoparticles will work for power delivery.

Reviewer #3 (Remarks to the Author):

The authors have addressed all the questions. I think it can be accepted for publication.

Communications Materials is committed to improving transparency in authorship. As part of our efforts in this direction, we are now requesting that all authors identified as 'corresponding author' create and link their Open Researcher and Contributor Identifier (ORCID) with their account on the Manuscript Tracking System prior to acceptance. ORCID helps the scientific community achieve unambiguous attribution of all scholarly contributions. You can create and link your ORCID from the home page of the Manuscript Tracking System by clicking on 'Modify my Springer Nature account' and following the instructions in the link below. Please also inform all co-authors that they can add their ORCID to their accounts and that they must do so prior to acceptance.

Version 2:

Decision Letter:

Dear Dr Baker,

Thank you for resubmitting your manuscript titled "Towards Muscle Replacement: Biocompatible Low-Voltage Electrothermal Actuators". I am delighted to say that we are happy, in principle, to publish a suitably revised version in Communications Materials.

We therefore invite you to edit your manuscript to comply with our journal policies and formatting style in order to maximise the accessibility and therefore the impact of your work.

EDITORIAL REQUESTS

* Your manuscript should comply with our policies and format requirements, detailed in our style and formatting guide (<https://www.nature.com/documents/commsj-phys-style-formatting-guide-accept.pdf>).

* Please edit your manuscript according to the editorial requests in the attached table, and outline revisions made in the right hand column. If you have any questions or concerns about any of our requests, please do not hesitate to contact me. It is important that each request be addressed in order to avoid delays in accepting your manuscript. Please upload the completed table with your manuscript files as a Related Manuscript file.

* The editorial requests table also includes a full list of the files that must be provided upon resubmission. Please upload your files according to this table.

OPEN ACCESS

Communications Materials is a fully open access journal. Articles are made freely accessible on publication. For further information about article processing charges, open access funding, and advice and support from Nature Research, please visit <https://www.nature.com/commsmat/open-access>

Please use the following link to submit your revised files:

Link Redacted

We hope to hear from you within two weeks; please let us know if the process may take longer.

Best regards,

Dr Jet-Sing Lee
Senior Editor
Communications Materials

**** Visit Nature Research's author and referees' website at www.nature.com/authors for information about policies, services and author benefits****

Reviewers' comments:

Reviewer #1 (Remarks to the Author):

The paper introduces the concept of biocompatible low-voltage electrothermal actuators and presents Bio35, a specific low-voltage (3.6 V) actuator along with experimental results. While the topic is indeed relevant and interesting to the field, there are several significant issues that, in my view, prevent the manuscript from meeting the rigorous standards of Communications Materials.

We thank the reviewer for the positive comments about the relevance of this topic and area of research. We have addressed all of their suggestions and concerns below:

To improve the paper, I offer the following suggestions:

1. The manuscript is poorly organized and lacks clarity. The introduction section is excessively long and disproportionate to the rest of the manuscript. A concise and focused introduction is crucial for setting the stage for the research, but this section contains too much background information and tangential details.

We have streamlined the introduction and shortened it by **25% (from 1137 words to 899)**, and carefully ensured that we present a concise and focussed introduction to set the stage for our detailed study. Rather than show the whole of the introduction here, we rather refer the editor to the marked-up manuscript where the **removed text is highlighted in red**.

2. The figures in the manuscript are of poor quality, with text and labels that are too small to be legible. The authors should ensure that all figures are of high resolution and that all text, labels, and axes are sufficiently large to be easily read by the journal's target audience.

We apologise for the quality of the images, due to incorrect online transfer during the submission process. This issue has been addressed now: all figures are provided in high resolution and the labels and text in all images normalised to provide consistency and ensure ease of viewing. Please note that Figure 2 was changed significantly and partly moved to the SI to ensure clarity:

Figure 2. a) T_g of epoxy cured layers vs percentage DDS used as a crosslinking agent, with the biological operational temperature range indicated by the light green bar; **b)** outer temperature profiles of **Bio35** actuator as a function of displacement; **c)** maximum displacement of the **Bio35** actuator upon potential application as a function of time; **d)** surface temperature as a function of time at different voltages where the actuator dimensions are $0.97 \times 60 \times 12.5$ mm (thickness \times length \times width). The plots show the average value of 3 repeats on 3 individual samples, as shown in the SI.

3. The format of the figure captions in the manuscript is inconsistent, which detracts from the professional appearance and readability of the article.

The inconsistencies in the figure captions have already been addressed in response to Point 2. We have however ensured that all figure captions were checked again and any other inconsistencies addressed throughout the manuscript.

4. The references in the manuscript do not adhere to the journal's formatting guidelines.

We have corrected all references to adhere to the guidelines.

Reviewer #2 (Remarks to the Author):

The manuscript presents an epoxy-based electrothermal actuator designed as a potential replacement for defective muscle caused by aging or disease. The proposed material utilizes a design strategy originating from work published in 1989, with modifications to optimize actuation under conditions relevant to biological applications. The study also includes cytotoxicity assessments and evaluates the actuator's potential to function as a valve for managing urinary incontinence using a mock urethral conduit. While the concept of employing a synthetic actuator for biological applications is intriguing, several key issues require further attention to improve the rigor and applicability of the work:

We thank the reviewer for their positive assessment of our work, also pointing to the long history of this approach, however without any biological application to date. We therefore appreciate the fact that the reviewer recognizes the significant strides we have made in real-life biological applications with our non-cytotoxic and biocompatible materials.

We have therefore addressed the reviewer's comments in detail below.

1. *The manuscript highlights the actuator's potential for managing urinary incontinence but overlooks alternative approaches that are already in use, such as pharmacological treatments, physical therapy, and injectable bulking agents. The introduction would benefit from a broader discussion of these strategies to better contextualize the proposed actuator's niche or advantages.*

We thank the reviewer for pointing to the broader background to managing especially female stress urinary incontinence (fSUI). We have added the following text to the valve applications sections to avoid increasing the introduction length (also keeping Reviewer 1's request in mind to streamline the introduction, and to allow us to address later reviewers' comments), highlighting the alternative approaches as background:

To realise the potential for **Bio35** to form sphincter-like replacement implantable valves (as part of a wider approach to address various potential solutions related to fSUI, where 1st and 2nd line treatments such as pelvic floor exercises and bulking agent injection fail) a simple valve using two actuators was prepared (see **Figure 6a**).

2. A critical consideration for implantation is the mechanism by which the actuator would function autonomously, given that natural valves operate under neural control. Additionally, practical details about the source and delivery of electrical power to the actuator need to be addressed.

We thank the reviewer for pointing this critical aspect of control and power deliver to our actuator, especially within an in vivo setting. We are very aware of these challenges, which we have now addressed as part of our conclusions, pointing to a whole new approach we are currently working on in our laboratories – remote control of actuation using embedded hyperthermic nanoparticles. This new approach, developed for in vivo use, will address the issues raised by the reviewer:

In this study we presented results from our investigations into the preparation and optimisation of Joule-heated soft actuators, with the long-term goal of developing fully biocompatible in vivo actuators to aid in addressing the challenge of fSUI. The thermal behaviour of the developed materials could be tuned by simple adjustment of the co-polymerisation ratios. Through simple processing and combination with an in-situ Joule heating configuration, we produced non-cytotoxic actuators that operate in the biological operational temperature range (37.2-42°C). We could furthermore demonstrate the opening and closing of an artificial sphincter-like arrangement with a simple valve based on our actuators. The sphincter approach presented here is only a first step towards addressing fSUI. Future work will focus on the development of sling-like implantable actuators to address wider descending bladder and fSUI issues, thus enabling better integration into current surgical procedures for addressing fSUI. **In addition, actuators with embedded hyperthermic nanoparticles materials will be explored for the development of remote controlled actuation, thus starting to address the important issues control and power deliver to these actuators.** ~~In addition,~~ Such solutions will also aim to alleviate the challenges of post-surgical adjustments that are currently unavailable to support patient recovery.

We also point out that neither the manufactured valves nor Bio35 in general, ~~actuate~~ under body temperature (37 °C) but require heating to the higher levels of biological operational temperature window (40-42 °C), which we show in Section 2.2 and Figure 2.

3. Actuator performance was evaluated over a relatively short duration (40 minutes), which is insufficient for applications requiring long-term reliability. Extended testing is necessary to assess durability and functional stability over a biologically relevant timeframe.

We apologise that our explanations did not present the total time under which we studied actuator performance. We tested operation of our actuators well over 40 minutes (200 cycles at 1 min heating, 1 min cooling totalling 400 minutes, with the actuators then used in subsequent studies, see S5-7). These facts are now noted and discussed more clearly in the manuscript (highlighted in green and yellow below) and SI:

To assess the applicability of the actuation system for longer-term use, the cyclability of the bilayers was first investigated. Displacement measurements showed that **Bio35** maintained consistent performance over 20 cycles (see **Figure 3a**) and **over 100 cycles (see SI, Figure S4)** using an applied voltage of 3.6 V, without significant loss of displacement. **With a typical frequency of 2 minutes per cycle, our actuator, in this initial study, already showed excellent stability for use over more than 400 minutes of repeat cycling.**

4. Figures and data presented in the manuscript should clarify the number of independent samples tested under each condition. If the results are based on a single sample, this should be explicitly stated, and the limitations should be acknowledged.

We thank the reviewer for pointing towards the important aspect of repeatability (and therefore wider applicability) in our approach; we apologise for this oversight in providing details of our testing regimes. We have added appropriate and available information for all samples tested and data sets presented throughout the manuscript. For example, the samples are shown as an average of 3 repeats on 3 individual samples as highlighted in the legend for Figure 2.

Figure 2. a) T_g of epoxy cured layers vs percentage DDS used as a crosslinking agent, with the biological operational temperature range indicated by the light green bar; **b)** outer temperature profiles of **Bio35** actuator as a function of time for each voltage applied; **c)** displacement of the **Bio35** actuator upon potential application as a function of time; **d)** surface temperature as a function of maximum displacement for different actuator thicknesses and **e)** isometric force from **Bio35** actuator as a function of the voltage applied, where the actuator dimensions are $0.97 \times 60 \times 12.5$ mm (thickness \times length \times width). The plots show the average value of 3 repeats on 3 individual samples, with details shown in the SI.

5. Although the authors argue that the working temperature of 42.7°C is safe, prolonged exposure to this level of heat can induce oxidative stress and structural damage in surrounding tissues. Additional experiments are needed to evaluate the effects of long-term heat exposure on adjacent biological structures.

We agree with the reviewer that prolonged exposure to this level of heat can induce damage to surrounding tissues. Although extensive studies related to such issues lie beyond the scope of our study, we have now performed additional cell viability tests on a working actuator system (cycling voltage and thus also operating temperature) for 10 cycles. This additional data shows no decrease in biological activity of the cells under conditions relevant to our initial operating parameters. However, there are some challenges with long-term use of this approach, which we acknowledge and see as challenges for future development. We have added the relevant text and an additional figure to the SI:

To test the effect of thermally driven actuation on toxicity, cultured Saos-2 cell monolayers in 24-well plates were exposed to a working Bio35 actuator after 24 h. The actuator was immersed in the wells to make direct contact with the cells. Two different voltages (3 V and 5 V) and two cycle counts (5 and 10), with each actuation cycle lasting 1 minute of heating 1 minute of cooling, were tested. A non-actuated cell monolayer served as the control. Following actuation, alamarBlue analysis revealed no significant difference (n.s.) was observed between the control group and the 3 V, 5-cycle group (**Figure S8**). However, an 8.7% decrease in cell metabolic activity with increasing voltage and cycle count (5V, 10 cycles, $p < 0.05$) was observed. These result highlights the validity of our initial approach, while also pointing towards the specific challenges for future development.

Figure S8. Effect of actuation on Saos-2 cells using two voltages (3 V and 5 V) and two cycle counts (5 and 10). Metabolic activity decreased significantly with increasing voltage and cycle number (* $p < 0.05$).

6. The cytotoxicity assessment was conducted over a relatively short duration (24 hrs). Testing over a longer period (e.g., at least one week) is necessary to account for potential material degradation and the release of toxic substances in a physiological environment.

We agree with the reviewer that cytotoxicity test should, ideally, be performed for longer than 24h. We have therefore, to ensure we showcase our scientifically sound approach and the wide applicability of our newly formulated polymeric actuators and their non-cytotoxic behaviour, tested our actuators for 10 days. These tests show the stability of our materials and continued non-cytotoxic behaviour. We have adjusted Figure 5 and the legend to show this data. We have also adjusted the SI to further complement these findings; see also our response to Point 5 above.

Figure 5. (a) Acute cytotoxicity testing was performed using an alamarBlue assay after Saos-2 cells were exposed to Bio35 for 24 h, 7 days and 10 days. There was no significance (n.s.) in the difference in metabolic activity between the Bio35 group and the control group (N = 6) observed in

any of the time points. (b) Confocal fluorescence microscopy of Saos-2 cells cultured on Bio35 Actuator for 24 h, fixed and stained for F-actin (phalloidin, green) and nuclei (DAPI, blue). No observable difference was observed in either the number or morphology of cells on the actuator, compared to the control of tissue culture plastic. Scale bar = 200 μm .

Reviewer #3 (Remarks to the Author):

This paper propose a bilayer electrothermal actuator for potential applications in urinary sphincter replacements. The produced non-cytotoxic actuators can operate under low voltage and in the biological operational temperature range (37.2-42°C). The materials are also proved biocompatible. The actuation performance and durability are also tested. Finally, a simple valve actuator is demonstrated to mimic the opening and closing of an artificial sphincter. Main comments are as follows:

We thank the reviewer for accurately summarising the main points from our paper that showcases the scientific progress and advantages of our approach. We have addressed the reviewer's comments below.

1. *The font size in figures seems too small.*

We have responded to this comment in the corrections made to Reviewer 1, Comments 2 and 3, and all issues related to legibility of fonts, legends and captions for all figures have been addressed.

2. *I don't think only 100 cycles can prove the good durability of the actuators, especially for potential biocompatible applications, the working environments (such as temperature, humidity, PH, etc.) can affect the applicability and durability. Please add some discussion about this.*

We have addressed this point in detail in response to Reviewer 2 Comment 3. We are aware that long-term testing of these developed actuators would be desirable, but such studies fall outside the scope of our investigation, and is part of on-going activities and further investigations in our laboratories.

3. *What's the resistance of this actuator? During actuation does the resistance vary with time or voltage?*

We thank the reviewer for highlight this omission of relevant information. The electrical resistance of the wire is given by the manufacture of $< 100 \Omega/\text{m}$. Monitoring 5 actuation cycles gave the resistance of our actuators as $75 \pm 1 \Omega/\text{m}$ when in the on state at full displacement. The text of the manuscript has been updated to reflect this:

To assess the applicability of the actuation system for longer-term use, the cyclability of the bilayers was first investigated. Displacement measurements showed that **Bio35** maintained consistent performance over 20 cycles (see **Figure 3a**) and over 100 cycles (see **SI, Figure S4**) using an applied voltage of 3.6 V, without significant loss of displacement (in each case resistance of our actuators was $75 \pm 1 \Omega/\text{m}$, monitored over 5 cycles, when in the on state at full displacement).

4. *What's the power or energy density of this actuator?*

We have added an additional line of text at the end of the paragraph under Figure 2 to provide information to the reader on the dimensions, power generated and energy density of our used Bio35 actuator:

The optimal thickness for **Bio35** (0.97 mm thickness, 60 mm length, and 12.5 mm width) generated a power of 18.36 mW with an energy density of 29.7 J/g.

5. For the valve application, actually the pressure in the reservoir has significant effect on the flow rate and also the actuation of the valve actuator, because we know the urinary bladder is elastic. The demonstration only show the free flow of the water by gravity. The applicability of this valve actuator is not convincing.

We thank the reviewer for pointing these issues out and apologise that we did not make the experimental setup and operating conditions clear to the reader from the start. We had the same reservations when designing the experiment and so included a steady drip of fluid into, as well as out of, the bladder to ensure that the reservoir was at a constant volume to keep the flow rate and pressure constant. We have updated the text to highlight this to the reader:

To realise the potential of **Bio35** to form sphincter-like replacement implantable valves (as part of a wider approach to address various potential solutions related to fSUI), a simple valve using two actuators was prepared (see **Figure 6a**). A mock urethral conduit was set up consisting of a reservoir of water (300 mL, kept at a constant volume [± 0.5 mL] via a drip feed monitored at all times to ensure constant pressure/flow rate) to mimic the bladder and a low-density polyethylene tube (ID = 6 mm, wall thickness = 1 mm, length 60 mm, **Figure 6b**) to mimic the urethral conduit (typically with ID = 6 mm, length 40 mm).

6. The actuation strain and force are both very small compared to many other electrothermal actuators, please give some reason or discuss about some potential strategy for improvement.

We agree with the reviewer that both strain and force involved in the operation of our actuators are very small. We would however like to highlight the application area that we have targeted, and therefore the suitability of our actuator in this specific application:

For this specific problem investigated (stress urinary incontinence), which poses significant challenges and major impact on health, wellbeing and the economy, we believe that the strain and force generated here is more than sufficient to deal with this specific challenge.

We have also added a further clinically relevant suggestion in the SI as to a way to increase the effectiveness of the actuator in this application after discussion with clinicians. We have added clear reference to this additional information in the main text to direct readers:

To increase the effectiveness of the actuator it is suggested that implantation will involve a sling like motif (see **SI Figure S8**).

Figure S8. Schematics showing a) where the proposed actuator would be placed to contribute to solutions to stress urinary incontinence; b) the proposed actuator attachment to a sling to give aid to reduce stress urinary incontinence.

17 June 2025

Thank you for providing the reviewers' comments on our paper, manuscript COMMSMAT-24-0781-T, entitled:

"Towards Muscle Replacement: Biocompatible Low-Voltage Electrothermal Actuators"

We greatly appreciate the opportunity to address the queries raised during the second review process and are grateful for the constructive feedback and valuable suggestions provided by the reviewers.

We have carefully revised the manuscript and sort editorial input to address the concerns of reviewer 1.

Below, we have therefore provided detailed responses to each of the reviewers' comments, along with our fully revised and updated manuscript for your consideration. Please note:

- our responses to each of the reviewers' comments (*In italics*) are given in blue.
- A marked-up version of the relevant parts of the manuscript is provided under each response, with all changes to the text highlighted in yellow, and deleted text highlighted in red.
- A version of the manuscript with no markup or highlighted text is also provided.
- We have also added an acknowledgement section to highlight the extra work carried out by M.V.H. to address reviewers concerns throughout the reviewing process

We are confident that these modifications effectively address all concerns raised and so bring our manuscript in alignment with the criteria for publication in *Communications Materials*.

With our best regards,

Ben, Charl and co-authors

Reviewers' comments:

Reviewer #1 (Remarks to the Author):

The authors have carefully addressed the review comments. But I think the current version is not suitable for acceptance. Below are detailed comments to guide revisions.

1. Compare and contrast the proposed Bio35 actuator with existing approaches to clearly highlight its novelty and advantages.

We point to an already extensive comparison in text in paragraphs 3-5 of the Introduction. Whilst we removed many of the words to shorten the paragraph in response to early requests, we have kept the number of references to increase impact and to highlight the novelty of the Bio35. The advantages are also clearly highlighted by Figure 4.

2. Improve the quality and clarity of the figures. Consider using color coding, annotations, and captions to make the figures more self-explanatory.

We have asked for editorial input as we are not sure what this comment is referring to. We have made changes to several of the images in response to other comments by this author and now assume that our efforts and editorial input have ensured that these issues have been resolved.

3. Although the manuscript details the synthesis process of Bio35, I find that the clarity of Figure 1a remains inadequate for precise interpretation.

We have adjusted the image to address this comment and the issue of clarity. We have separated part (a) of the figure to show Bio35 and actuator preparation separately. Please see the updated figure below.

4. The camera perspectives in Figures 1c and 1d appear inconsistent, which may affect the reproducibility and interpretability of the experimental setup depicted.

We appreciate the reviewer's concern about reproducibility of our experimental work, and the importance of clarity and consistency in presenting results. Although the camera perspectives were consistent, images were slightly distorted due to refraction from the setup being placed in water. We have now added a fully annotated schematic to the SI (Figure S3) to clearly show the setup we used for these investigations, thus ensuring reproducibility and interpretability.

Figure S3. Experimental set up for the images obtained in **Figure 1 b** and **c** and **Videos S1** and **S2**.

The reader is alerted to this in the legend for Figure 1 of the article:

Figure 1: Formulation, manufacturing and application of low voltage and biological operational temperature range bilayer actuators: where **a**) schematic mechanism of bilayer actuator formation (specifically **Bio35**) from starting materials; **b**) configuration of the conductive thread Madeira HC 12 (100% polyamide/silver plated $<100 \Omega/m$) based thermal actuators; **c**) demonstration of gripper actuator formed from two bilayer actuators in atmospheric conditions and **d**) submerged in phosphate-buffered saline (PBS) solution, where the mass of the actuator is 1 g and the on/off voltage is 5V (200 mA). Please note: the experimental set up is detailed in **Figure S3**.

5. The numerical labels and annotations in Figure 5a are too small.

We thank the reviewer for alerting us to this issue. We have now adjusted the image as follow:

6. The schematic presented in Figure 6a does not adequately depict the electromechanical actuation process of the valve actuator.

The Figure is now updated to fully clarify the actuator mechanism and applied voltage-dependent behaviour:

7. The manuscript presents data on the displacement and force generated by Bio35 actuators at 3.6 V. However, it would be useful to include data on the response time of the actuators and the relaxation time.

We would like to highlight that the information is already present in the SI; however, we have now mentioned this additional information explicitly in the main text to direct the reader to the SI:

The response time of the actuators (< 2 seconds after applied voltage) and the relaxation time (< 30 sec after voltage is removed) are shown in **Figure 3** and **Figure S5 and 6**.

8. It would be helpful to provide more information about the design of the mock urethral conduit, the specific parameters used during the valve opening and closing tests (e.g., flow rate, pressure).

Please see our comments to Reviewer 3, Comment 5 from the last round of reviews, presented here for your convenience:

5. For the valve application, actually the pressure in the reservoir has significant effect on the flow rate and also the actuation of the valve actuator, because we know the urinary bladder is elastic. The demonstration only show the free flow of the water by gravity. The applicability of this valve actuator is not convincing.

We thank the reviewer for pointing these issues out and apologise that we did not make the experimental setup and operating conditions clear to the reader from the start. We had the same reservations when designing the experiment and so included a steady drip of fluid into, as well as out of, the bladder to ensure that the reservoir was at a constant volume to keep the flow rate and pressure constant. We have updated the text to highlight this to the reader:

To realise the potential of **Bio35** to form sphincter-like replacement implantable valves (as part of a wider approach to address various potential solutions related to fSUI), a simple valve using two actuators was prepared (see **Figure 6a**). A mock urethral conduit was set up consisting of a reservoir of water (300 mL, kept at a constant volume [± 0.5 mL] via a drip feed monitored at all times to ensure constant pressure and flow rate) to mimic the bladder and a low-density polyethylene tube (ID = 6 mm, wall thickness = 1 mm, length 60 mm, **Figure 6b**) to mimic the urethral conduit (typically with ID = 6 mm, length 40 mm).

9. Extend the duration and scope of the biocompatibility tests to ensure the long-term safety and stability of the Bio35 material in vivo.

We point out that this study is already groundbreaking in nature, demonstrating the first biocompatible system of this kind (i.e., novelty). We have already performed initial tests to demonstrate biocompatibility and provided significant extension to our earlier tests after initial review. Any further work now falls without the scope of this study and publication. Other reviewers (especially Reviewer 2, who requested significant additional bio-related studies) are fully satisfied of the novelty and level of our experimental work, and recommended publication.

10. The references section is comprehensive, but some key citations could be updated to include more recent studies on artificial muscles and soft robotics.

We have reviewed the literature, but we cannot find the references the reviewer is alluded to. As the other reviewers have only highlighted the use of nanoparticles as a missing reference, we assume that this is now resolved (see response to Reviewer 2).

Reviewer #2 (Remarks to the Author):

The authors made sincere efforts to respond to the original comments. In my opinion, the manuscript is quite qualified for acceptance.

However, there is one suggestion on the response to "A critical consideration for implantation is the mechanism by which the actuator would function autonomously, given that natural valves operate under neural control. Additionally, practical details about the source and delivery of electrical power to the actuator need to be addressed." The proposed hyperthermia nanoparticles as a power delivery measure are speculative. If particular published works support this statement, authors should cite the papers. Or, authors should elaborate on how nanoparticles will work for power delivery.

We thank the reviewer for their kind comments. We have addressed the concern above by adding the additional relevant References 31 and 48,49 and 50 to the text, highlighted below:

Artificial muscles can be designed to respond to various stimuli, including electrical signals²⁶, thermal changes²⁷, and pneumatic pressure^{4,20}. These stimuli can be used to induce different types of motion as a result of the contraction and relaxation of the materials²⁸. Electrothermal actuators have shown significant promise owing to their precise control and rapid responses^{29,30,31}.

In addition, actuators with embedded hyperthermic nanoparticles materials will be explored for the development of remote controlled actuation, thus starting to address the important issues control and power deliver to these actuators^{48,49,50}. Such solutions will also aim to alleviate the challenges of post-surgical adjustments that are currently unavailable to support patient recovery.

31. Chen M. Advanced Soft Electronics in Biomedical Engineering, 2024.

48. Lin X, Han M. Recent progress in soft electronics and robotics based on magnetic nanomaterials. Soft Science 2023, 3(2).

49. Wang B, Shen J, Huang C, Ye Z, He J, Wu X, et al. Magnetically driven biohybrid blood hydrogel fibres for personalized intracranial tumour therapy under fluoroscopic tracking. Nat Biomed Eng 2025.

50. Kwon H, Yang Y, Kim G, Gim D, Ha M. Anisotropy in magnetic materials for sensors and actuators in soft robotic systems. Nanoscale 2024, 16(14): 6778-6819.

Reviewer #3 (Remarks to the Author):

The authors have addressed all the questions. I think it can be accepted for publication.

We thank the reviewer for acknowledging that we have addressed all raised queries, and that our study is now suitable for publication.